# Sacubitril/Valsartan in Heart Failure with Reduced Ejection Fraction: Real-World Experience from Italy (the REAL.IT Study)

**DOI:** 10.3390/jcm12020699

**Published:** 2023-01-16

**Authors:** Andrea Di Lenarda, Gabriele Di Gesaro, Filippo Maria Sarullo, Daniela Miani, Mauro Driussi, Michele Correale, Claudio Bilato, Andrea Passantino, Erberto Carluccio, Alessandra Villani, Luca degli Esposti, Chiara d’Agostino, Elena Peruzzi, Simone Poli, Massimo Iacoviello

**Affiliations:** 1Cardiovascular Center, University Hospital and Health Services of Trieste, 34128 Trieste, Italy; 2U.O. Cardiologia IRCCS ISMETT, 90133 Palermo, Italy; 3U.O.S. Di Riabilitazione Cardiovascolare Ospedale Buccheri La Ferla Fatebenefratelli, 90123 Palermo, Italy; 4SOC Cardiologia, Dipartimento Cardiotoracico, Azienda Sanitaria Universitaria Friuli Centrale, Ospedale S. Maria della Misericordia, 33100 Udine, Italy; 5SC Universitaria di Cardiologia AOU “Ospedali Riuniti”, 71122 Foggia, Italy; 6U.O.C. Cardiologia Azienda ULSS 8 Berica—Ospedali dell’Ovest Vicentino, 36071 Arzignano, Italy; 7Division of Cardiology and Cardiac Rehabilitation, U.O. Cardiologia ICS Maugeri SpA SB Bari, IRCCS Istituto di Bari, 70124 Bari, Italy; 8Cardiologia e Fisiopatologia Cardiovascolare, Azienda Ospedaliera Universitaria “Santa Maria della Misericordia”, 06156 Perugia, Italy; 9U.O. Day Hospital—MAC Cardiologia, Istituto Auxologico Italiano—Ospedale S. Luca, 20149 Milan, Italy; 10CliCon S.r.l. Società Benefit, 40137 Bologna, Italy; 11Cardio-Metabolic Medical Manager, Novartis Farma SpA, 20154 Milan, Italy; 12Evidence Generation & Data Analytics Head, Novartis Farma SpA, 20154 Milan, Italy; 13RWE Data Analyst, Novartis Farma SpA, 20154 Milan, Italy; 14Surgical and Medical Sciences Department, University of Foggia, 71122 Foggia, Italy

**Keywords:** angiotensin receptor-neprilysin inhibitor, heart failure with reduced ejection fraction, NYHA functional class, real-world practice, sacubitril/valsartan

## Abstract

Sacubitril/valsartan reduces heart failure (HF)-related hospitalizations and cardiovascular mortality in PARADIGM-HF and has become a foundational treatment for HF with reduced ejection fraction (HFrEF). However, data of its routine real-world use are limited, and evidence from Italian settings is lacking. The REAL.IT study aimed to characterize the demographics, pharmacotherapy, clinical characteristics and outcomes of sacubitril/valsartan-treated Italian patients with HFrEF. Electronic medical records of patients initiating sacubitril/valsartan from October 2016 to June 2019 at nine specialized hospital outpatient HF centers across Italy were reviewed. Overall, 924 adults (mean age 64.5 years, 84.6% male) were included. At baseline, 38.7% had an ischemic HF etiology, 45.9% hypertension, 23.2% atrial fibrillation, 25.4% diabetes mellitus, 26.1% an implantable cardioverter-defibrillator and 31.9% coronary artery bypass grafting. There were no clear patterns of patient selection over time. During follow-up, NYHA class improved in 37.5% of patients after a mean of 5.3 ± 3.8 months; 36.1% and 16.7% of patients were in NYHA class III during characterization and after one year of follow-up, respectively. Left ventricular ejection fraction (LVEF) improved ≥5% in 56.3% of patients at one year; 39.7% had ≥30% reduction of N-terminal pro-B-type natriuretic peptide; 2.2% had hyperkalemia during characterization and 2.6% during follow-up; and 3.8% had hypotension during characterization and 12% during follow-up. A total of 50 (5.8%) of patients had device implantation (ICD/CRT) during follow-up. HF-related hospitalization was recorded in 19.6% of patients during follow-up; 3.8% of patients died, approximately 1.3% from cardiovascular causes. Our real-world data confirm the favorable effectiveness and tolerability of sacubitril/valsartan observed in pivotal randomized controlled trials.

## 1. Introduction

Over the last decades, the epidemiology of heart failure (HF) has been described as a global pandemic [1]. In Italy, HF prevalence has been estimated to be around 1.7% (i.e., in excess of 1 million people) and increases sharply with age [2,3]. Despite the progress of therapeutic approaches and the availability of comprehensive clinical practice guidelines, HF is still characterized by high morbidity and mortality [4,5]. Moreover, the underuse and underdosing of the recommended disease-modifying drugs has been shown to contribute to an even worse prognosis [6].

In this setting, real-world evidence reflecting the broader and unselected population encountered in routine clinical practice can describe patient characteristics and treatment patterns that differ from and complement those reported in randomized clinical trials which generally have more restrictive inclusion criteria. This evidence is even more relevant when analyzing the most recent drugs which have demonstrated their favorable effects, such as the first-in-class angiotensin receptor neprilysin inhibitor (ARNI) sacubitril/valsartan. In fact, although sacubitril/valsartan has been established as a foundational treatment for HF based on clinical trial data, there remains a paucity of data from real-world settings about its use, the characteristics of patients treated in routine practice, their resource utilization and the effect on outcomes. Furthermore, evidence from clinical experience of sacubitril/valsartan in Italian settings is limited [7]. Therefore, observational studies using data from patients already treated with sacubitril/valsartan are needed to evaluate the use of this therapy in the real-world setting of everyday clinical practice in Italy, considering the indication criteria provided by the therapeutic plan.

To address this research gap, we undertook a multicenter, retrospective cohort study using data from Italian outpatient specialist clinics to describe the characteristics of patients treated with sacubitril/valsartan, including baseline demographics, pharmacotherapy, clinical characteristics, outcomes and resource and drug utilization. The principle focus of this paper is to describe the baseline demographic, clinical characteristics and clinical outcomes of Italian patients with HFrEF; an analysis of pharmacotherapy and resource and drug utilization will be reported later.

## 2. Patients and Methods

### 2.1. Study Design

This was an observational, retrospective, non-interventional cohort study based on electronic medical records from nine specialized hospital outpatient heart failure centers geographically distributed in the national territory of Italy. The principal investigators of REAL.IT.

### 2.2. Study Population

Consecutive ambulatory patients with a diagnosis of HF that attended the outpatient clinics for HF management and who had been prescribed sacubitril/valsartan from 1 October 2016 (the year of its launch in Italy) to 30 June 2019 (the inclusion period) were included in the study. The 30 June 2019 cut-off date allowed for a follow-up period of at least one year at the time of data extraction (30 June 2020). Specifically, all patients attending one of the outpatient clinics for the diagnosis and treatment of HF in the Italian centers involved who were ≥18 years old and had at least one prescription of sacubitril/valsartan from 1 October 2016 to 30 June 2019 were included in the study. The index date was defined as the date of the first prescription of sacubitril/valsartan during the inclusion period. A characterization period, i.e., the 6-month period before the index date, was used to assess the baseline characteristics of the overall patient population. The analyses comparing characterization and follow-up period were performed on patients with available information on the variables of interest during both periods. Patients with missing age or sex information were excluded from the analysis. The study design is shown in Figure 1.

### 2.3. Data Collection

Retrospective data were collected using the electronic medical records and the administrative databases of the centers, integrating patients’ personal health records, health examination records and hospital visit records. The study was conducted in compliance with the Declaration of Helsinki and current Italian regulations for observational studies. The ethics committee of each participating center was notified of the study, and the ethics committee of each participating center approved the study.

### 2.4. Variables

Patients were characterized using the following variables, with data collected no earlier than 6 months before the index date (i.e., the characterization period): demographic and previous clinical history, etiology of HF, presence of comorbidities, NYHA functional class, systolic blood pressure (SBP), hematochemical laboratory tests, including brain (B-type) natriuretic peptide (BNP) or N-terminal (NT)-proBNP (NT-proBNP), chronic kidney disease (CKD), electro- and echocardiographic data, cardiopulmonary exercise testing and pharmacological treatments for HF.

After 12 months of follow-up, the following variables were re-evaluated: clinical and functional parameters, including NYHA class, SBP, heart rate (HR), symptoms and signs of HF, routine hematochemical laboratory tests, standard electrocardiogram (ECG), and 2-dimensional (2-D) echocardiography to assess cardiac capacity and the effect of sacubitril/valsartan on cardiac function indices.

To assess the severity of HF, the following events were examined during follow-up: hospitalization related to HF, other CV events, non-CV causes, receipt of implantable cardioverter-defibrillator/coronary artery bypass grafting (PCI/CABG), valvular intervention, device implantation (ICD/cardiac resynchronization therapy [CRT]), number of emergency room visits and CV and non-CV mortality. As the study was based on the secondary use of data, safety monitoring and safety reporting were not a focus.

### 2.5. Objectives

The primary objective of the REAL.IT (real-world experience in HFrEF patients treated with sAc/vaL in Italy) study was to describe the characteristics of the patients treated with sacubitril/valsartan, including baseline demographics, pharmacotherapy and clinical characteristics in an Italian specialist setting overall and by calendar quarter (relating to time since launch).

A full listing of secondary and explorative objectives is provided in Appendix A. Objectives relevant to this paper are as follows: to evaluate, overall and by calendar quarter (relating to time since launch), the frequency of CV and non-CV death and the effect of treatment with sacubitril/valsartan on cardiac function indices.

### 2.6. Statistical Analysis

Continuous data were summarized in terms of the number of observations, the number of missing values, mean, and standard deviation (SD). Categorical data were summarized regarding the number of patients providing data, with frequency counts and percentages of patients in each category. Percentages were calculated using the number of observations with non-missing values as the denominator. The Chi-square test was used for categorical variables and the paired *t*-test was used to detect differences between the characterization and the follow-up periods for continuous variables (systolic and diastolic blood pressure, heart rate, and ejection fraction). Statistical significance was set at *p* < 0.05. All statistical analyses were performed using STATA SE, version 12.0 (StataCorp LLC, College Station, TX, USA). Data management was carried out using Microsoft SQL Server 2012.

## 3. Results

### 3.1. Patient Disposition

A total of 948 HF patients with at least one prescription of sacubitril/valsartan during all data availability periods were identified among the nine centers involved in the study. Of these, 924/948 adult HF patients were included in the analyses as they had at least one prescription of sacubitril/valsartan during the inclusion period, were without missing age or sex data and had 6 months of characterization period and at least 12 months of follow-up available. A flow chart of the study is shown in Figure 2.

Analyses comparing the characterization and follow-up periods were performed on patients with available information on the variables of interest during both periods. To maximize the value of the data within the database, further analyses were also carried out on all patients with available data for each variable considered.

Patient demographic and clinical characteristics are presented in Table 1. The mean age of patients was 64.5 ± 11.9 years and the majority of patients were in the age range 60–69 years (32.3%), followed by 70–79 years (27.2%) and 50–59 years (20.0%).

Seven-hundred and eighty-two (84.6%) patients were male, and 38.7% had an ischemic etiology of HF. A total of 45.9% of patients had hypertension, 25.4% diabetes mellitus, 31.9% PCI/CABG and 23.2% atrial fibrillation (Table 1).

Hypotension was recorded for 21 (3.8%) of patients during the characterization period, and 2.2% of patients had hyperkalemia. A total of 225 (26.1%) of patients had device implantation (ICD/CRT) during the characterization period.

When baseline data were stratified by calendar quarter, there were no clear patterns of selection of patients over time (Appendix A). Patients in each calendar quarter showed similar characteristics, and few differences were reported. Mean age in the three most recent quarters tended to be slightly older than in earlier quarters; the duration of HF disease tended to be slightly longer and symptoms more severe in the first quarters. However, there were no consistent trends in clinical and demographic characteristics over time.

A sub-analysis of baseline data in patients with 80% available data (*N* = 538) showed that 43.5% had hypertension, 35.3% had ischemic heart disease, 21.2% had diabetes mellitus and 28.3% PCI/CABG. In patients with 80% available data, 81.4% had a 2-D echocardiogram during the characterization period and 41.9% during follow-up.

As shown in Table 2, a significant reduction in the prescription of diuretics was observed during follow-up, whereas no differences were found in the prescription of mineralocorticoid receptor antagonists (MRAs) and beta-blockers.

### 3.2. Impact on NYHA Class

Baseline characteristics according to NYHA functional class showed that 63.8% were in NYHA class II during the characterization period and 66.4% during follow-up (Table 2). During the characterization period, 36.1% of patients were in NYHA class III. During follow-up, the proportion of patients in NYHA class III had reduced to 16.7%, and 16.2% (91 patients) were in class 1. NYHA class was unchanged in 57.4% of patients after a mean of 5.3 ± 3.8 months of follow-up and improved in 37.5%. NYHA class worsened in just 5.1% of patients after a mean of 6.3 ± 3.9 months follow-up (*p* < 0.001).

### 3.3. Impact on Ejection Fraction

Of the 316 patients with available data in both the characterization and follow-up periods, mean EF increased from 28.7% to 32.8% (*p* < 0.001) (Table 2). Of these patients, the percentage of patients with an improvement of left ventricular ejection fraction (LVEF) ≥5% was 56.3% (178/316 patients); 40 (12.7%) of patients had a worsening of LVEF of ≥5%, and 98 (31.0%) had LVEF values stable between ± 5% after a mean of 5.4 ± 4.1 months.

### 3.4. Hospitalizations and Events during Follow-Up

Among patients with available data (*N* = 744), 19.6% had at least one HF-related hospitalization during follow-up (Table 3). This represents 0.2 HF-related hospitalizations per patient during the one year of follow-up. Thirty-five (3.8%) patients died during follow-up, of which, according to the data availability, at least 12 were from CV causes.

Fifty patients (5.6%) of the 863 patients had device implantation during follow-up, without evidence during the characterization period.

Hypotension, the most common adverse event reported with sacubitril/valsartan in the PARADIGM-HF trial [8], was recorded for 66 patients (12.0%) during follow-up (Table 2).

### 3.5. Laboratory Data

Routine laboratory tests during characterization and follow-up periods are reported in Table 4. Of particular interest, mean values of NT-proBNP decreased by 20% during follow-up (from 1970 ng/L to 1580 ng/L, *p* < 0.001) among the 412 patients with available data. Specifically, 39.7% (164/413) and 27.3% (15/55) of patients had ≥30% reduction of NT-proBNP and BNP, respectively; 47.0% (194/413) and 36.4% (20/55) had stable values of NT-proBNP and BNP (±30%), respectively, while 13.3% (55/413) and 36.4% (20/55) had an increase of ≥30% of NT-proBNP and BNP values, respectively. A mild worsening of kidney function was reflected in a downward trend of around 5% in estimated glomerular filtration rate (eGFR) (Table 4). The proportion of patients with hyperkalemia was 2.6% during follow-up (Table 4).

## 4. Discussion

This present real-world, multicentric study provides a thorough characterization of patients who started therapy with sacubitril/valsartan in settings of routine clinical practice in Italy. The data from electronic medical records of HF patients referred to nine centers geographically distributed throughout the Italian national territory give insights into the baseline findings and changes in clinical characteristics during the first year of treatment with sacubitril/valsartan. Overall, a significant proportion of patients had improvements in EF, NYHA class, and levels of natriuretic peptides in the year after initiation of sacubitril/valsartan. Rates of HF-related hospitalization and death were low.

The demographic and clinical characteristics of the 924 patients included in the study were broadly in line with other international and real-world studies of patients treated with sacubitril/valsartan. When compared with the population of PARADIGM-HF, our population was characterized by a similar mean age of patients, (64.5 years in REAL.IT, compared with 63.8 years in the PARADIGM-HF) [8]. Further, the patients were younger than those of European and international real-world studies on sacubitril/valsartan utilization, whose ages ranged between 66 and 70 years [7,9,10,11,12]. Moreover, when compared with PARADIGM-HF, a lower percentage of patients were affected by diabetes (25.4% vs. 34.7%) and atrial fibrillation (23.2% vs. 36.2%). Interestingly, in our series a greater percentage of patients were taking diuretics and MRA when compared with PARADIGM-HF (87 vs. 80% and 77 vs. 56%). These differences are similar to those observed in other real-world studies [9]. It should be noted that patients in REAL.IT were referred to highly specialized centers for HF treatment and, therefore, may have been better controlled by cardiologists initiating sacubitril/valsartan therapy more promptly than may otherwise have occurred. The most frequent comorbidities encountered in the REAL.IT study were hypertension (45.9%) and ischemic heart disease (38.7%). Hypertension is also the most commonly reported comorbidity in real-world studies of sacubitril/valsartan [7,9,12], and the prevalence in our cohort falls in the middle of the reported range. Rates of both hypertension and ischemic heart disease etiology are lower in our study than in PARADIGM-HF [8], and may have been underestimated in REAL.IT perhaps partly because of the challenges of collecting real-world data. Moreover, the reported ischemic heart disease etiology of HF in sacubitril/valsartan real-world studies varies widely, ranging from 25% to 82% in systematic reviews [4,9].

NYHA stratification at baseline (64% NYHA II and 36% NYHA III) was consistent with other real-world data, which reports around 63% of patients with NYHA II and 30% with NYHA III [9]. Overall, a significant proportion of patients improved their clinical characteristics after starting therapy with sacubitril/valsartan. More than a third (37.5%) of patients moved to a less symptomatic NYHA class after around 5 months. One year after starting sacubitril/valsartan, the proportion of patients with NYHA III decreased from 36.1% to 16.7%, while the proportion of patients with NYHA I, absent during the characterization period, increased to 16.2%. Similar results were obtained in the Ariadne Registry, with almost 50% of patients improving from NYHA III to NYHA II and 16–18% from NYHA II to NYHA I [13]. This effect was also documented in the systematic review of nine studies by Proudfoot et al. [9] and by Haddad et al. [14].

In parallel, among patients with available data, 56.3% had an increased LVEF of ≥5% during follow-up, and 39.7% of patients had a ≥30% reduction in NT-proBNP levels after starting sacubitril/valsartan therapy. Elevated levels of natriuretic peptides are helpful in assessing prognosis in HF, and the statistically significant reduction in natriuretic peptides after starting sacubitril/valsartan is of particular relevance. Reductions in NT-proBNP concentrations were associated with sacubitril/valsartan-related benefits in the PARADIGM-HF study [8], and have been shown to be associated with reverse left ventricular remodeling during other guideline-directed medical therapies for HF [15,16]. The favorable effects of sacubitril/valsartan in reversing cardiac remodeling and improving systolic function as well as their relationship with NT-proBNP reduction have been also observed in a recent exploratory study at 12 months [17]. It was hypothesized that the reverse cardiac remodeling, established to be associated with improved prognosis in HF [18] as seen in our study, might offer a mechanistic explanation for the effects of sacubitril-valsartan in patients with HFrEF [17]. Other real-world studies have also examined levels of natriuretic peptides, showing consistent patterns of reduction in NT-proBNP in clinical practice [9,17]. In REAL.IT, the mean EF at baseline was 28.7%, which is within the range reported in the literature, between 23% and 38% [9]. Similarly to other published studies [19,20,21,22,23,24], there were indications of cardiac reverse remodeling in our patients following initiation of sacubitril/valsartan (LVEF increased ≥5% in 56.3%), and over 60% of patients had improvements in EF after a mean time of approximately 6 months. In PARADIGM-HF, sacubitril/valsartan was shown to be superior to enalapril in reducing hospitalizations for worsening HF, as well as CV-related and all-cause mortality in patients with ambulatory HFrEF [8]; additionally, and real-world comparative studies have consistently shown that sacubitril/valsartan reduces the rate of HF-related hospitalizations compared with standard of care [9]. In REAL.IT, 19.6% of patients had hospitalizations for HF in the year after sacubitril/valsartan initiation, a value within the range reported in other studies analyzing HF-related hospitalizations after the initiation of sacubitril/valsartan [7,9,13]. Accurate identification of the cause of death can be more problematic in a real-world setting than in more tightly controlled, randomized controlled trials; in REAL.IT, all-cause mortality during a year of follow-up was 3.8% (35 patients), although CV causes were confirmed for only approximately a third. Given the potential under-reporting of the outcomes analyzed, these data and the low proportion of deaths observed during the first year of follow-up could suggest effective management and a protective effect of sacubitril/valsartan in patients with HFrEF. Of interest, two studies that compared mortality in patients treated with sacubitril/valsartan or ACEi/ARB reported a rate of all-cause death of 6.8% at 6 months with sacubitril-valsartan versus 34.1% with ACEi/ARB [25], and 8.9% versus 12.2% (*p* < 0.05) [11].

Hypotension, as has been reported in other trials of sacubitril/valsartan, was the most common adverse event reported [8,26]. Patients who experienced a hypotensive episode, regardless of treatment assignment, were more likely to be on sub-target doses of ACEi or ARB before screening. Moreover, despite the higher incidence of symptomatic hypotension in sacubitril/valsartan-treated patients in PARADIGM-HF, the number of patients who had to discontinue the study drug was higher in the enalapril group, and sacubitril/valsartan recipients who developed hypotension also derived similar clinical benefits to those who did not experience hypotension, even if down-titrated to a lower dose of sacubitril/valsartan [27].

A mild worsening of kidney function was observed during follow-up, with a downward trend in the estimated eGFR for around 5%, which is consistent with the literature [7]. However, hyperkalemia was stable before and after starting therapy, with a minimal proportion of patients with hyperkalaemia on treatment. Of interest, concerns about the effects of ARNI on kidney function have been addressed in a recent cohort study of 27,166 patients, that found sacubitril/valsartan was not associated with higher rates of acute kidney injury than renin-angiotensin system inhibitors in patients newly intiating therapy [28]. Indeed, hyperkalemia and a decline of eGFR have been reported to be less likely to occur during treatment with sacubitril/valsartan than conventional ACEi therapy [29,30]. Additionally, as seen in this study, the use of sacubitril/valsartan may reduce the requirement for diuretics [25,31,32].

As this was a retrospective observational study analyzing anonymized data derived from electronic medical records, the study shares some limitations inherent in this type of study. For example, primary care data were not collected. Therefore, it was not possible to evaluate information about the role of primary care physicians in managing patients. In addition, the available data was incomplete for some patients, although this was accounted for in the analyses. Specifically, events can be underestimated if they are not meticulously registered in electronic records. Patients in the study were referred to the nine highly specialized HF centers, where they were managed by cardiologists that could have initiated the start of sacubitril/valsartan treatment more promptly than may otherwise have occurred. Therefore, they may not be fully representative of patients seen in routine general practice, and our findings may not be generalizable to the general population.

## 5. Conclusions

These real-world data on patients with HF treated with sacubitril/valsartan in settings of daily clinical practice in Italy confirm the effectiveness and tolerability observed within previous pivotal randomized clinical trials.

## Figures and Tables

**Figure 1 jcm-12-00699-f001:**
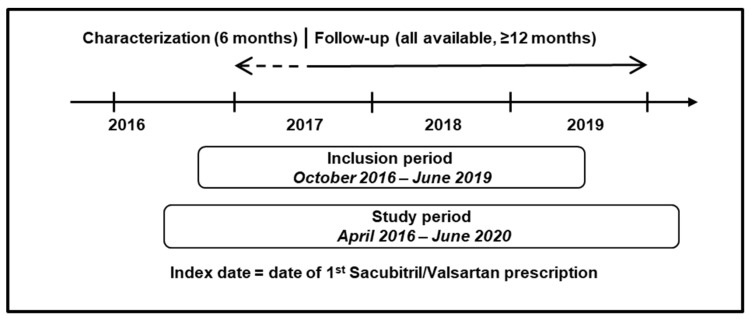
Study design. The study period was between 1 April 2016 and 30 June 2020, and the inclusion period was between 1 October 2016 and 30 June 2019. There was a characterization period of 6 months before the index date (the date of the first prescription of sacubitril/valsartan), and all patients were followed up for at least 12 months from the index date to June 2020.

**Figure 2 jcm-12-00699-f002:**
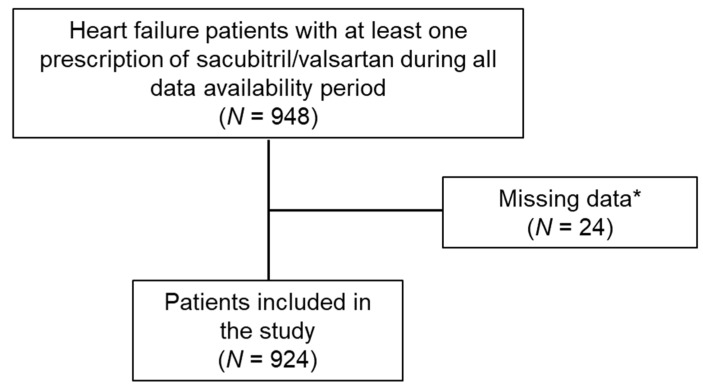
Flow chart of the study. * Patients with missing age or sex data and/or who did not have 6 months of characterization period and at least 12 months of follow-up available were excluded.

**Table 1 jcm-12-00699-t001:** Baseline demographic and clinical characteristics of patients treated with sacubitril/valsartan during the characterization period *.

Characteristic	
	*N* = 924
Age, (years) mean ± SD	64.5 ± 11.9
Gender, *n* (%)	
Male	782/924 (84.6)
Female	142/924 (15.4)
Ischemic heart disease, *n* (%)	232/599 (38.7)
PCI/CABG, *n* (%)	191/599 (31.9)
Moderate or severe mitral or aortic valvular disease, *n* (%)	99/599 (16.5)
Implanted prosthetic valve, *n* (%)	33/229 (14.4)
Devices (ICD/CRT), *n* (%)	225/863 (26.1)
Atrial fibrillation, *n* (%)	214/924 (23.2)
Prior hospitalization for HF, *n* (%)	194/538 (36.1)
Previous stroke, *n* (%)	17/460 (3.7)
Diabetes mellitus, *n* (%)	235/924 (25.4)
Hypertension, *n* (%)	424/924 (45.9)
Chronic kidney disease, *n* (%)	142/597 (23.8)
CKD ^§^, *n* (%)	304/829 (36.7)
Duration of HF disease ** (years), mean ± SD	6.8 ± 6.0

* Data are shown for patients with available data during the characterization period, which was defined as the 6-month period before the index date, the date of first prescription of sacubitril/valsartan during the inclusion period. Patients were followed up for at least 12 months. ** *N* = 580 patients with available data. ^§^ considering patients with available data for diagnosis of CDK and those for whom CDK was deduced from creatinine value considering the eGFR. Abbreviations: CABG = coronary artery bypass grafting; CKD = chronic kidney disease; CRT = cardiac resynchronization therapy; HF = heart failure; ICD = implantable cardioverter defibrillator; PCI = percutaneous coronary intervention; SD = standard deviation.

**Table 2 jcm-12-00699-t002:** Clinical examinations in patients with available data during characterization and follow-up periods *.

Characteristic	Patients, *n*	Characterization Period	Follow-Up Period	*p* Value
NYHA Class, *n* (%)				
I	563	NI ^a^	91 (16.2)	<0.001
II	563	359 (63.8)	374 (66.4)	
III	563	203 (36.1)	94 (16.7)	
IV	563	–	4 (0.7)	
Blood pressure, mmHg, mean ± SD				
SBP	551	121 ± 16	114 (17)	<0.001
DBP	551	73 ± 10	68 (10)	<0.001
Hypotension **, *n* (%)	551	21 (3.8)	66 (12.0)	<0.001
Heart rate, BPM, mean ± SD	455	69 ± 11	67 ± 10	<0.001
Electrocardiogram, *n* (%)	727	570 (78.4)	417 (57.4)	<0.001
2-D Echocardiogram, *n* (%)	924	752 (81.4)	387 (41.9)	<0.001
Ejection fraction, %, mean ± SD	316	28.7 ± 5.8	32.8 ± 7.9	<0.001
Drug therapy				
ACE-i, *n* (%)	686	462 (67.3)	27 (3.9)	<0.001
ARBs, *n* (%)	686	185 (27.0)	20 (2.9)	<0.001
Beta-blockers, *n* (%)	553	533 (96.4)	531 (96.0)	0.753
MRA, *n* (%)	553	425 (76.9)	425 (76.9)	1
Diuretics, *n* (%)	686	599 (87.3)	566 (82.5)	0.013
Ivabradine, *n* (%)	686	51 (7.4)	58 (8.5)	0.485
Digitalis, *n* (%)	686	36 (5.2)	41 (6.0)	0.558

* The characterization period was defined as the 6-month period before the index date, the date of the first prescription of sacubitril/valsartan during the inclusion period. Patients were followed up for at least 12 months. ** Hypotension was defined as SBP < 95 mmHg. ^a^ Data available for only ≤3 patients were not included (NI) in the analysis. Abbreviations: 2-D = 2-dimensional; ACE-i: angiotensin-converting enzyme inhibitor; ARBs: angiotensin II receptor blockers; BPM = beats per minute; DBP = diastolic blood pressure; MRA: mineralocorticoid receptor antagonists; NYHA = New York Heart Association; SBP = systolic blood pressure; SD = standard deviation.

**Table 3 jcm-12-00699-t003:** Cardiovascular and unrelated events during follow-up *.

Parameter	Patients with Available Data, *n*	Follow-Up Period
Hospitalization		
HF-related, *n* (%)	744	146 (19.6)
Per patient during the 1-year period of follow-up, mean ± SD	744	0.2 ± 0.5
Other CV events, *n* (%)	579	118 (20.4)
Per patient during the 1-year period of follow-up, mean ± SD	579	0.2 ± 0.4
Non-CV causes, *n* (%)	461	89 (19.3)
Per patient during the 1-year period of follow-up, mean ± SD	461	0.2 ± 0.5
Device implantation (ICD/CRT), *n* (%)	863	50 (5.8)
HF-related emergency room visits, *n* (%)	333	66 (19.8)
Death **, *n* (%)	924	35 (3.8)

* Patients were followed up for at least 12 months. ** 16 patients had the information regarding the CV-related cause of death. CRT = cardiac resynchronization therapy; CV = cardiovascular; HF = heart failure; ICD = implantable cardioverter-defibrillator; SD = standard deviation.

**Table 4 jcm-12-00699-t004:** Routine laboratory test results in patients with available data during characterization and follow-up periods *.

Characteristic	Patients with Available Data, *n*	Characterization Period	Follow-Up Period	*p* Value
Hemoglobin (g/dL), mean ± SD	347	13.6 ± 1.6	13.5 ± 2.0	0.480
Glycated hemoglobin (mmol/mol), mean ± SD	87	50 ± 14	49 ± 15	0.445
Creatinine (mg/dL), mean ± SD	551	1.20 ± 0.35	1.28 ± 0.48	<0.001
Estimated glomerular filtration rate **, mean ± SD	551	68.06 ± 22.51	64.97 ± 22.94	<0.001
45.00 ≤ eGFR < 60.00), *n* (%)	551	114 (20.7)	126 (22.9)	<0.001
30.00 ≤ eGFR < 45.00), *n* (%)	551	87 (15.8)	96 (17.4)	<0.001
15.00 ≤ eGFR < 30.00), *n* (%)	551	8 (1.5)	22 (4.0)	<0.001
eGFR < 15.00), *n* (%)	551	–	NI ^a^	<0.001
Sodium (mmol/L), mean ± SD	503	140 ± 5	141 ± 3	0.008
Potassium (mmol/L), mean ± SD	507	4.65 ± 6.13	4.43 ± 0.50	0.427
Hyperkalemia ^†^, *n* (%)	507	11 (2.2)	13 (2.6)	0.679
B-type natriuretic peptide (ng/L), mean ± SD	55	424 ± 354	504 ± 783	0.491
NT-proBNP (ng/L), mean ± SD	413	1970 ± 2650	1580 ± 2633	<0.001
Uric acid (mg/dL), mean ± SD	51	6.16 ± 1.86	6.04 ± 1.79	0.565

* The characterization period was defined as the 6-month period before the index date, the date of the first prescription of sacubitril/valsartan during the inclusion period. Patients were followed up for at least 12 months. ** calculated by the CKD-EPI method. ^†^ Defined as K > 5.4 mmol/L. ^a^ Data available for ≤3 patients only were not included (NI) in the analysis. Abbreviations: BNP = B-type natriuretic peptide; eGFR = estimated glomerular filtration rate; NT = N-terminal; SD = standard deviation.

## Data Availability

Data are available from the corresponding author upon reasonable request.

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
