# Peer review of "Sacubitril/Valsartan in Heart Failure with Reduced Ejection Fraction: Real-World Experience from Italy (the REAL.IT Study)"

_jcm, 2023, doi:10.3390/jcm12020699_

Round 1

Reviewer 1 Report

The paper focuses on analysis of data collected in real world patients treated with ARNI.

The topic attracts interest, but in the paper some major flaws deserve to be mended.

The baseline HF and medical therapy in place at time of study patient enrollment is not reported as well as the average ARNI dose that was in place at time of patient follow up.

The missing data prevent any consideration on the relevance of declared achieved benefit on limited LVEF increase and on NT proBNP  decrease

A point per point analysis on similarities and differences  between REAL.IT study population and PARADIGM HF study population is also advised, as for instance  in the PARADIGM HF treated arm, atrial fibrillation was present in 36% of patients and diabetes in 34,6%, vs 23,2% and 24,5% in the current observational study.

Author Response

We would like to thank the reviewer for his/her helpful comments.

The paper focuses on analysis of data collected in real world patients treated with ARNI. The topic attracts interest, but in the paper some major flaws deserve to be mended. The baseline HF and medical therapy in place at time of study patient enrollment is not reported as well as the average ARNI dose that was in place at time of patient follow up.

Response: Thank you for the very relevant point raised. We have now added the data in Table 2. We have also commented them in the results section and in discussion section.

The missing data prevent any consideration on the relevance of declared achieved benefit on limited LVEF increase and on NT proBNP decrease

Response: We hope that reporting the missed data could help to clarify the relevance of the observed benefit.

A point per point analysis on similarities and differences between REAL.IT study population and PARADIGM HF study population is also advised, as for instance in the PARADIGM HF treated arm, atrial fibrillation was present in 36% of patients and diabetes in 34,6%, vs 23,2% and 24,5% in the current observational study.

Response: We reported the comparison with the population of REAL.IT and of the main observational studies in the discussion section (lines 279-288):

“When compared with the population of PARADIGM-HF, our population was characterized by a similar mean age of patients, (64.5 years in REAL.IT, compared with 63.8 years in the PARADIGM-HF) [8]. Further, the patients were younger than those of European and international real-world studies on sacubitril/valsartan utilization which ranged between 66 and 70 years [7,9-12]. Moreover, when compared with PARA-DIGM-HF a lower percentage of patients was affected by diabetes (25.4% vs. 34.7%) and atrial fibrillation (23.2% vs. 36.2%). Interestingly, in our series a greater percentage of patients were taking diuretics and MRA when compared with PARADIGM-HF (87 vs. 80% and 77 vs. 56%). These differences are similar to those observed in the other real-world studies [9].”

Reviewer 2 Report

Dear Author(s),

Real-world evidence is highly valuable to generalize RCT trial findings to real-world settings, elucidate treatment effect heterogeneity - factual efectiveness and safety profile, and identify potential rare ADRs.

Your present manuscript is well-written, methodology is adequate, results are clearly presented and conclusions are supported by the results.

However, I have several minor comments/suggestions:

a) Discussion can be improved. More detail focus should be put on comparison with other RWE body of literature.

b) Would you rather use the term effectiveness instead of efficacy, since you are talking about RWE findings?

c) If I were you, I would paraphrase the sentence “A recent exploratory study has now shown that a reduction in the concentration of NT-proBNP was weakly yet significantly correlated with improvement in markers of cardiac volume and function at 12 months (p<0.001) following initiation of sacubitril/valsartan [15], strongly suggesting that treatment of HFrEF with sacubitril/valsartan may improve cardiac structure and/or function.” … Since this was only an observation you can only talk about plausible association, but not about causality (latter only from RCT findings). What is more, due to only a weak association you should not talk about a ‘strong’ suggestion.

Author Response

We would like to thank the reviewer for his/her helpful comments.

Real-world evidence is highly valuable to generalize RCT trial findings to real-world settings, elucidate treatment effect heterogeneity - factual efectiveness and safety profile, and identify potential rare ADRs. Your present manuscript is well-written, methodology is adequate, results are clearly presented and conclusions are supported by the results.

Response: Thank you for having appreciated our manuscript.

However, I have several minor comments/suggestions:

  1. a) Discussion can be improved. More detail focus should be put on comparison with other RWE body of literature.

Response: Thank you for the suggestion. We added in the discussion some information and comparison with other relevant RWE papers at Lines 307-310:

Similar results were obtained in Ariadne Registry with almost 50% of patients improving from NYHA III to NYHA II and 16-18% from NYHA II to NYHA I [13],  and also documented in the systematic review of 9 studies shown by Proudfoot et al [9] and by Haddad et al [14].

And at lines 331-333 :

Interestingly, in the multicenter Italian registry DISCOVER-ARNI [24] 60% of patients with LVEF ≤35% or NYHA II-III at baseline, improved and lost the indication to ICD implantation after 6 months of treatment with sacubitril/valsartan.

  1. b) Would you rather use the term effectiveness instead of efficacy, since you are talking about RWE findings?

Response: Thank you, we’ve changed throughout the manuscript.

  1. c) If I were you, I would paraphrase the sentence “A recent exploratory study has now shown that a reduction in the concentration of NT-proBNP was weakly yet significantly correlated with improvement in markers of cardiac volume and function at 12 months (p<0.001) following initiation of sacubitril/valsartan [15], strongly suggesting that treatment of HFrEF with sacubitril/valsartan may improve cardiac structure and/or function.” … Since this was only an observation you can only talk about plausible association, but not about causality (latter only from RCT findings). What is more, due to only a weak association you should not talk about a ‘strong’ suggestion.

Response: Thank you for the comment, we’ve changed the sentence as follows at lines 324-329:

“The favorable effects of sacubitril/valsartan in reversing cardiac remodeling and improving systolic function as well as their relationship with NT-proBNP reduction have been also observed in a recent exploratory study at 12 months [17].”

Round 2

Reviewer 1 Report

This real world study has limited data collection in the investigated population and LVEF was detected only in 1/3 of subjects. Also for this reason It doesn't seem appropriate to cite the DISCOVER-ARNI study result as  appropriate to confirm the ARNI efficacy in reverting  left ventricular remodeling.

The DISCOVER-ARNI study was based on a tiny population (#113) and the proportion of subject with ischemic etiology in the subgroup requiring ICD was 51% vs 35%, downgrading the relevance of any statistical assessment.

Author Response

We would like to thank the reviewer for this further helpful comment. This is our reply.

This real world study has limited data collection in the investigated population and LVEF was detected only in 1/3 of subjects. Also for this reason It doesn't seem appropriate to cite the DISCOVER-ARNI study result as  appropriate to confirm the ARNI efficacy in reverting  left ventricular remodeling.

The DISCOVER-ARNI study was based on a tiny population (#113) and the proportion of subject with ischemic etiology in the subgroup requiring ICD was 51% vs 35%, downgrading the relevance of any statistical assessment.

Response

We have removed the sentence describing in detail the results of DISCOVER-ARNI ("Interestingly, in the multicenter Italian registry DISCOVER-ARNI [24] 60% of patients with LVEF ≤35% or NYHA II-III at baseline, improved and lost the indication to ICD implantation after 6 months of treatment with sacubitril/valsartan.").